# Crosstalk between Mast Cells and Lung Fibroblasts Is Modified by Alveolar Extracellular Matrix and Influences Epithelial Migration

**DOI:** 10.3390/ijms22020506

**Published:** 2021-01-06

**Authors:** Mariam Bagher, Oskar Rosmark, Linda Elowsson Rendin, Annika Nybom, Sebastian Wasserstrom, Catharina Müller, Xiao-Hong Zhou, Göran Dellgren, Oskar Hallgren, Leif Bjermer, Anna-Karin Larsson-Callerfelt, Gunilla Westergren-Thorsson

**Affiliations:** 1Unit of Lung Biology, Department of Experimental Medical Sciences, Lund University, 221 84 Lund, Sweden; Mariam.Bagher@gmail.com (M.B.); Oskar.Rosmark@med.lu.se (O.R.); Linda.Elowsson_Rendin@med.lu.se (L.E.R.); Annika.Nybom@med.lu.se (A.N.); catharinam@gmx.de (C.M.); gunilla.westergren-thorsson@med.lu.se (G.W.-T.); 2Department of Respiratory Medicine and Allergology, Skåne University Hospital, Lund University, 221 85 Lund, Sweden; Oskar.Hallgren@med.lu.se (O.H.); leif.bjermer@med.lu.se (L.B.); 3Lund University Bioimaging Centre, Lund University, 221 85 Lund, Sweden; Sebastian.Wasserstrom@med.lu.se; 4Bioscience Department, Respiratory, Inflammation and Autoimmunity, IMED Biotech Unit, AstraZeneca, 431 53 Mölndal, Sweden; Xiao-Hong.Zhou@astrazeneca.com; 5Department of Cardiothoracic Surgery and Transplant Institute, Sahlgrenska University Hospital, 413 45 Gothenburg, Sweden; goran.dellgren@vgregion.se

**Keywords:** lung fibroblasts, mast cells, epithelial cells, extracellular matrix, IL-6, tryptase, vascular endothelial growth factor, hepatocyte growth factor, idiopathic pulmonary fibrosis

## Abstract

Mast cells play an important role in asthma, however, the interactions between mast cells, fibroblasts and epithelial cells in idiopathic pulmonary fibrosis (IPF) are less known. The objectives were to investigate the effect of mast cells on fibroblast activity and migration of epithelial cells. Lung fibroblasts from IPF patients and healthy individuals were co-cultured with LAD2 mast cells or stimulated with the proteases tryptase and chymase. Human lung fibroblasts and mast cells were cultured on cell culture plastic plates or decellularized human lung tissue (scaffolds) to create a more physiological milieu by providing an alveolar extracellular matrix. Released mediators were analyzed and evaluated for effects on epithelial cell migration. Tryptase increased vascular endothelial growth factor (VEGF) release from fibroblasts, whereas co-culture with mast cells increased IL-6 and hepatocyte growth factor (HGF). Culture in scaffolds increased the release of VEGF compared to culture on plastic. Migration of epithelial cells was reduced by IL-6, while HGF and conditioned media from scaffold cultures promoted migration. In conclusion, mast cells and tryptase increased fibroblast release of mediators that influenced epithelial migration. These data indicate a role of mast cells and tryptase in the interplay between fibroblasts, epithelial cells and the alveolar extracellular matrix in health and lung disease.

## 1. Introduction

Fibroblasts are major producers of extracellular matrix (ECM) and have a key role in dysregulated lung function and remodeling processes in chronic lung diseases such as asthma, chronic obstructive pulmonary disease (COPD) and idiopathic pulmonary fibrosis (IPF) [1,2,3,4]. IPF is a lethal interstitial lung disease with poor prognosis. The disease is histologically characterized by heterogeneous areas of subpleural fibrosis, epithelial damage in the alveolar region, fibroblast foci and accumulation of ECM [5,6]. Disruptions in the processes of injury and wound healing starting with recurring alveolar epithelial micro injuries are suggested to drive abnormal tissue remodeling, with activated inflammatory cells, synthesis of ECM and release of proinflammatory mediators [7,8,9]. Involved mediators in lung fibrosis, such as interleukin 6 (IL-6) [10], vascular endothelial growth factor (VEGF) [11] and hepatocyte growth factor (HGF) [12], are produced by fibroblasts and may induce both pro- and antifibrotic events in the alveolar regions. It has long been known that mast cell numbers are elevated in lungs from IPF patients [13,14], and a positive correlation between increased mast cell numbers and myofibroblast accumulation was found in lung tissue explants from IPF patients [15]. Mast cells produce a cocktail of different cytokines, proteases and ECM components that are involved in fibrotic diseases and may drive fibroblast differentiation [16,17,18]. However, despite evidence that mast cells may activate fibroblasts and contribute to remodeling, the mechanism for this interaction is poorly understood. We have recently shown that mast cells and mast cell tryptase promote fibroblast migration via protease-activated receptor 2 (PAR2) [19]. To follow up on these results, we evaluated the effect of mast cells and mast cell proteases tryptase and chymase on the release of mediators from lung fibroblasts derived from healthy individuals and IPF patients. We then wanted to further study the interaction between fibroblasts and mast cells by investigating how the microenvironment affects the crosstalk between these cells and the subsequent effect on migratory capacity of epithelial cells. In the present study, we used human decellularized lung tissue slices (lung scaffolds) with remaining alveolar ECM, developed in our laboratory as a 3D cell culturing model [20], in order to observe if an ECM context of the alveolar regions alters the interaction of fibroblasts and mast cells and the release of mediators. Our previous studies show that the ECM microenvironment has a fundamental impact on fibroblast activity. Fibroblasts cultured in decellularized lung scaffolds synthesized a different ECM profile compared to cells cultured on traditional cell culture plastic plates [20]. More interestingly, fibroblasts cultured in a fibrotic ECM microenvironment directed healthy fibroblasts to synthesize a fibrotic ECM profile compared to fibroblasts cultured in a healthy ECM microenvironment [21]. Our new data indicate that the interaction between fibroblasts and mast cells is modulated when cultured in a 3D matrix environment compared to conventional 2D cell cultures on plastic plates. In addition, we demonstrate that mast cells and mast cell tryptase induce altered cytokine and growth factor profiles from healthy and IPF fibroblasts which affect the migration of alveolar epithelial cells.

## 2. Results

### 2.1. Altered Mediator Profile in Fibroblasts Co-Cultured with Mast Cells on Plastic Plates vs. 3D Lung Scaffolds

As we have previously shown that lung fibroblasts are highly responsive to their ECM context, we wanted to investigate if the interaction between fibroblasts and mast cells would be different when cultured in 3D lung scaffolds compared to conventional 2D cell cultures on plastic plates. We analyzed the release of IL-6, VEGF and HGF by enzyme-linked immunosorbent assay (ELISA) in the two culture conditions using human fetal lung fibroblasts (HFL-1) and LAD2 mast cells (Figure 1a–f). The LAD2 cells in these experiments were not stimulated to induce degranulation. Mast cells alone released low amounts of IL-6 (12.62 ± 1.62 pg/mL) from cells cultured on plastic plates and there were no detectable levels of IL-6 in scaffold cultures. Fibroblasts were the major producers of IL-6. There were trends towards increased IL-6 release in co-cultures with mast cells both on traditional cell culture plastic plates (*p* = 0.082) and in scaffold cultures (*p* = 0.083) compared to fibroblasts in monoculture (Figure 1g). Mast cells in monoculture released low amounts of HGF (19.19 pg/mL) in cell culture plastic plates and in scaffold cultures (30.61 ± 1.59 pg/mL). Fibroblasts were the main producers of HGF and the release of HGF was significantly increased when fibroblasts were co-cultured with mast cells on plastic culture plates (*p* = 0.037) compared to fibroblasts alone, in contrast to scaffold cultures, where no significant differences were observed (Figure 1h). Mast cells in monoculture did not release detectable levels of VEGF, whereas low amounts of VEGF were released from the mast cells in scaffold cultures (16.85 ± 1.59 pg/mL). Fibroblasts were the major producers of VEGF. However, there were no significant effects on VEGF release in co-cultures with fibroblasts and mast cells in either culture condition. Interestingly, there were increased levels of VEGF from fibroblasts cultured in scaffolds compared to cells cultured on plastic plates (*p* = 0.035) (Figure 1i). These results imply that the ECM microenvironment of the alveolar compartment influences the interaction between fibroblasts and LAD2 mast cells that may differ between growth factors and cytokines.

### 2.2. Mast Cells and Mast Cell Proteases Alter Mediator Profile in Healthy and IPF-Derived Lung Fibroblasts

We further wanted to study the differences in activity between lung fibroblasts derived from IPF patients and healthy individuals. Lung fibroblasts from patients with IPF and healthy individuals were co-cultured with mast cells or stimulated with mast cell tryptase and/or chymase in order to mimic the MC_TC_ mast cell subtype, which contains both tryptase and chymase. There were increased IL-6 levels in co-cultures with mast cells compared to monocultures in healthy fibroblasts (*p* = 0.05) (Figure 2a). The levels of secreted VEGF were about doubled in healthy fibroblasts compared to IPF fibroblasts (*p* = 0.017). VEGF levels tended to be higher in co-cultures with mast cells in both healthy and IPF fibroblasts (Figure 2b). HGF synthesis tended to be higher in IPF-derived fibroblasts compared to healthy both in monocultures and in co-cultures with mast cells (Figure 2c). Stimulation with tryptase alone, or in combination with chymase increased the release of VEGF and especially HGF in both healthy and IPF fibroblasts (Figure 2d–f). In contrast, chymase did not induce any further release of either IL-6 or VEGF (Figure 2d,e). Interestingly, stimulation with chymase alone appeared instead to decrease HGF synthesis in both healthy and IPF-derived lung fibroblasts (Figure 2f). Mast cells in monoculture released IL-6 (2.08 ± 0.17 pg/mL), VEGF (89.39 ± 3.5 pg/mL) and no detectable levels of HGF in the multiplex analyses.

### 2.3. Migratory Capacity and Metabolic Activity of Fibroblasts are Affected by Mast Cell Proteases

To study the effect of tryptase and chymase on fibroblast migration, tryptase (75 ng/mL) and chymase (1 ng/mL) were added to HFL-1 cells. In line with our previous data [19], tryptase significantly enhanced the migratory capacity of fibroblasts at 24 h (*p* = 0.001), 48 h (*p* = 0.001) and 72 h (*p* = 0.001), whereas chymase did not show any significant effect on fibroblast migration (Figure 3a). Higher concentration of chymase (10 and 100 ng/mL) appeared to be toxic, where the cells detached from the cell culture plastic plate and migration could not be observed (data not shown). The promigratory effect of mast cell tryptase was reduced in combination with chymase, indicating an antimigratory effect of chymase on fibroblasts (Figure 3a). Stimulations with either tryptase (75 ng/mL), chymase (1 ng/mL) or the combination of tryptase and chymase did not show any effect on metabolic activity in HFL-1 (Figure 3b).

### 2.4. Effects on α-SMA Expression after Co-Culture with Mast Cells

α-*smooth muscle actin* (SMA) expression in lung fibroblasts was quantified by Western blot analysis. There were no significant differences in α-SMA levels between fibroblasts in monocultures compared to co-cultures with LAD2 cells. There were no significant differences in α-SMA levels between fibroblasts and co-cultures in the presence of the PAR2 inhibitor P2pal-18s, although there was a trend (*p* = 0.079) towards reduced α-SMA expression in co-culture with LAD2 cells. As expected, stimulation with the positive control TGF-β (10 ng/mL) significantly increased α-SMA expression in fibroblasts (*p* = 0.042) and in co-cultures (*p* = 0.038) (Figure 4a,b).

### 2.5. Migratory Effects on Epithelial Cells are Affected by HGF and IL-6 and Conditioned Media from HFL-1

To mimic crosstalk between mast cells, fibroblasts and epithelial cells in the alveolar region, the migratory effects induced by IL-6, VEGF and HGF were evaluated on A549 epithelial cells. Furthermore, the migratory effect induced by conditioned media from fibroblasts in monoculture and co-cultures with mast cells in lung scaffolds or cell culture plastic plates were evaluated on the epithelial cells. The concentrations of each mediator were based on the levels measured in the different culture conditions (as presented in Figure 1). IL-6 at the different concentrations 0.3, 1 and 3 ng/mL significantly reduced migration of the epithelial cells at different time points, 12 h (*p* < 0.05; *p* < 0.001) and 24 h (*p* < 0.01 and *p* < 0.001) (Figure 5a), whereas HGF significantly stimulated migration at both 5 and 10 ng/mL at 12 h (*p* < 0.001) and 24 h (*p* < 0.05 and 0.001) (Figure 5b). VEGF did not have a significant effect on epithelial migration (Figure 5c). Conditioned media from HFL-1 cells cultured alone or in co-culture with unstimulated LAD2 mast cells for 72 h in the two different culture systems, plastic plates and lung scaffolds, both significantly increased the migration of epithelial cells compared to epithelial cells without any stimuli (controls) at the different time points. However, the culture conditions induced differing effects on the migration of A549, where medium collected from fibroblasts cultured in scaffolds with and without mast cells induced faster migration of the epithelial cells at 6 h (*p* < 0.05), 12 h (*p* < 0.001) and 24 h (*p* < 0.01) compared to medium from fibroblasts cultured with and without mast cells on plastic culture plates (Figure 5d).

## 3. Discussion

In the current study, we investigated the effect of mast cells and mast cell proteases on mediator release from fibroblasts and if cell–cell interactions were modified when a natural lung ECM environment was introduced. We could observe an altered secretion of IL-6, VEGF and HGF in co-culture with mast cells or after stimuli with tryptase in fibroblasts obtained from healthy individuals and patients with IPF. We further investigated the effect on cell migration and could confirm that tryptase induced fibroblast migration. Mediator release differed between the 3D lung scaffold model compared to traditional 2D cell culture plastic plates, suggesting that the ECM structure in the lung scaffold had an important role in the release and storage of cytokines and growth factors. We could further show that epithelial migration was stimulated by HGF and inhibited by IL-6.

Several studies have reported that mast cells and fibroblasts colocalize in the lung [14,15,16,22], where mast cell granules containing merely tryptase and not chymase are described to be the dominating mast cell type in the alveolar regions [15], and that mast cells are located close to fibrotic foci and alveolar type II cells in lung tissue samples from IPF patients [22]. Lung fibroblasts are also known to have an altered phenotype in the diseased lung of IPF patients [23,24]. In line with these findings, mast cells and tryptase may have profibrotic properties by promoting both recruitment of fibroblasts and myofibroblast differentiation in fibrosis-related diseases [25,26,27,28,29], which our data on fibroblast migration support. However, we could not establish an induced myofibroblast phenotype by mast cells measured by Western blot as alterations in α-SMA levels in fibroblasts cultured on plastic plates, or an inhibitory effect by the PAR2 antagonist. This could be due to the presence of both chymase and tryptase in LAD2 cells ([30]. We have previously shown that mast cells and tryptase enhanced the migratory capacity of fibroblasts, which was attenuated by a PAR2 antagonist, confirming that the mechanism behind the induced migration of fibroblasts involve PAR2 activation [19], as supported by other studies [22]. In the present study, tryptase enhanced fibroblast migration. On the contrary, chymase did not affect fibroblast migration and the combination of chymase and tryptase reduced the promigratory effect of tryptase. Chymase is an important protease in both pathological and physiological processes in the lung [31]. Chymase has previously been shown to have an antifibrotic role in renal fibrosis [32] and on airway smooth muscle cell function by degrading matrix components [33]. In the present study, we followed up the results obtained on fibroblast migration and investigated whether mast cells and mast cell proteases altered mediator release from fibroblasts. Interestingly, the mast cell proteases tryptase and chymase appeared to have opposite effects on mediator release from healthy and IPF fibroblasts. Tryptase increased the release of IL-6, VEGF and HGF in healthy fibroblasts with less response in IPF fibroblasts, whereas chymase instead appeared to have an inhibitory effect on the release of VEGF and especially HGF. However, due to the limited availability of patient material in this study, we cannot draw any further conclusions and these data are warranted to be followed up.

The presence of mast cells in co-cultures with fibroblasts promoted the release of proinflammatory IL-6 in primary lung fibroblasts obtained from healthy individuals and tendencies towards increased IL-6 in our different culture systems in lung scaffolds and plastic plates. A recent study showed that lung fibroblasts from IPF patients synthesized increased levels of IL-6 and that IL-6 receptors were overexpressed [10]. However, IL-6 appears to have dual roles depending on whether it is an acute or chronic situation, as IL-6 was crucial for lung repair in an influenza-induced lung injury model through reducing fibroblast accumulation and promoting epithelial cell survival [34].

In the present study, we also analyzed VEGF, an angiogenetic factor important for vascularization, which may also promote vascular remodeling [35]. The role of VEGF is, however, controversial in IPF. Reduced levels of VEGF-A in bronchoalveolar lavage (BAL) fluid from patients with IPF compared to control subjects have been observed in several studies [36]. However, the anti- or profibrotic effect of VEGF-A may depend on which isoforms are generated [37]. VEGF-A, secreted especially by epithelial type II cells, but also by fibroblasts and mast cells in the alveolar regions, has been highlighted to play a role in alveolar wall protection in pulmonary fibrosis [11]. In addition, we did observe reduced synthesis of VEGF-A in fibroblasts obtained from IPF patients compared to control subjects and increased VEGF-A synthesis after co-culture with mast cells or stimulation with tryptase. Furthermore, there were elevated levels of VEGF-A released in the lung scaffolds compared to the cell cultures on plastic plates, indicating that the 3D microenvironment has an impact on cellular activity.

HGF is another growth factor produced by fibroblasts that has been suggested to have antiapoptotic effects on epithelial and endothelial cells [38]. In vivo studies have reported that HGF induces apoptosis and prevents accumulation of myofibroblasts in experimental lung fibrosis, while in vitro studies have shown that HGF intercepts with SMAD signaling activity involved in the regulation of TGF-β and inhibits epithelial-mesenchymal transitions [12]. Interestingly, our data supports the notion that these discrepancies in the literature may be due to differences in experimental conditions regarding the ECM milieu. We did observe, although small, but a significant increase in HGF release from fibroblasts induced by co-culturing with mast cells, but when fibroblasts were co-cultured with mast cells in lung scaffolds, this effect did not appear, further highlighting the role which the ECM microenvironment has on cellular behavior. Data from our previously published proteomic study indicated that the HGF receptor was significantly decreased in lung tissue from IPF patients compared to control subjects [39].

In the present study, we investigated if cellular contact between mast cells and fibroblasts altered fibroblast activity by release of different mediators. Importantly, the mast cells were not activated in terms of degranulation to secrete inflammatory mediators and we did not detect any major release of the investigated mediators VEGF, IL-6 or HGF from the mast cells. Mast cells are otherwise well-known to release both IL-6 and VEGF upon activation and degranulation [40]. To further investigate the crosstalk between fibroblasts and mast cells during inflammation processes, future studies with activated/degranulated mast cells would be of interest to elucidate potential differences between stimulated and unstimulated mast cells. In our previous study by Bagher et al., Ig-E-activated mast cells did not further induce or alter the migration of fibroblasts compared to unstimulated mast cells [19]. Altogether, these data support that fibroblasts were the major producers of the measured mediators which were further increased in cellular contact with mast cells.

As reciprocal signaling between fibroblasts and alveolar epithelial cells are thought to be important for the development of IPF, we analyzed the effect of released mediators on the migratory capacity of epithelial cells. From our migration experiments on epithelial cells, it appears that HGF has a dose-dependent stimulatory effect and IL-6 an opposing inhibitory effect. The promigratory effect of HGF has previously been shown in other in vitro systems where HGF induced epithelial proliferation and migration [38]. However, the in vivo relevance of these and previous in vitro data may be questioned, as the stimulation of increased HGF release by mast cells was attenuated by the introduction of a complex alveolar ECM microenvironment in our 3D scaffold cultures. These results highlight the need of relevant culture systems, as we have previously shown that the lung scaffold milieu is directing the fibroblast phenotype [20,21].

The alveolar ECM milieu of the lung may thereby have an important role in regulating the activity of fibroblasts and their interactions with other cells such as mast cells, which was reflected in our 3D lung scaffold model. This may be due to matrix components binding and sequestering cytokines and growth factors as well as by direct interactions with cells that may modulate the responses. This study is limited by the availability of patient material which resulted in low *n*-numbers and that cell lines had to be used instead to follow up our results in a complex cell culture model. The cell line HFL-1 is a fetal lung fibroblast which may give different results compared to the primary adult fibroblasts obtained from healthy individuals or patients, and may account for some of the differences we observed in mediator release. However, our data obtained from both primary lung fibroblasts and HFL-1 are in line with other results presented within this research field, supporting a strong correlation between cellular behavior and culture conditions, but more extensive studies are needed.

In conclusion, our observations suggest that mast cells affect fibroblast function, including mediator release and cellular crosstalk with epithelial cells, indicating a potential role of these cells in crosstalk with fibroblasts in health and disease (Figure 6). The interactions between these cells are modulated by the ECM microenvironment, highlighting the importance of advancing work with more complex culture systems such as the 3D lung scaffold to increase the in vivo relevance of cell physiology studies in vitro. Further studies are therefore highly warranted to clarify the mechanisms for mast cell interactions with fibroblasts, which may provide novel strategies for preventing or treating uncontrolled tissue remodeling and lung fibrosis.

## 4. Materials and Methods

### 4.1. Biological Material

LAD2 mast cells (Dr. Arnold Kirshenbaum, Laboratory of Allergic Diseases, NIAID, Bethesda, MD, USA) were cultured in StemPro^®^-34 Serum-free medium with Stem Pro-34 nutrient supplement (Invitrogen, Waltham, MA, USA), 1% penicillin–streptomycin (PEST), 1% L-glutamine (both from Gibco BRL, Paisley, UK) and 100 ng/mL Stem cell factor (SCF) (Peprotech, Stockholm, Sweden). The epithelial cell line A549 (ATCC^®^ CCL-185™) was cultured in DMEM (Sigma-Aldrich, St Louis, MO, USA), supplemented with 10% fetal clone serum (FCIII, Thermo Scientific, Waltham, MA, USA), 50 µg/mL Gentamicin (Sigma-Aldrich, St Louis, MO, USA), 2.5 µg/mL Amphotericin B (Gibco BRL, Paisley, UK) and 1% Glutamax (Gibco, BRL, Paisley, UK). Primary lung fibroblasts were isolated from explant lungs from 3 healthy donors and 3 patients diagnosed with IPF as previously described [21]. The age of the patients was between 44 and 63 years. The primary lung fibroblasts were used between passages 4 and 7. The primary cell line human fetal lung fibroblasts (HFL-1; ATCC, Rockville, MD, USA) were used between passages 17 and 20. Fibroblasts were cultured in DMEM (Sigma-Aldrich, St Louis, MO, USA) supplemented with 10% fetal clone serum (FCIII, Thermo Scientific, Waltham, MA, USA), 1% PEST and 2 mM L-glutamine. Lung explant tissue from healthy organ donors with no history of lung disease were included. The healthy donor lungs were to be used for transplantation but were included in this study when no matching recipient could be identified.

### 4.2. Cell Culture Systems

Primary lung fibroblasts from healthy individuals and patients with IPF or HFL-1 were seeded in 6-well plastic cell culture dishes, in mono- or co-cultures with LAD2 mast cells, respectively. Seeding densities for primary fibroblasts were 0.13 × 10^6^ cells/well and LAD2-cells 0.1 × 10^6^ cells/well in the co-cultures. Initial culture was performed in DMEM with 10% FCIII, 1% PEST and 1% 2 mM L-glutamine.

### 4.3. Decellularization and Repopulation of Human Lung Scaffolds

Lung scaffolds were prepared as previously described [20]. Briefly, small cubic pieces, approximately 1 cm^3^, were dissected from healthy lung tissue. The tissue was snap frozen in 2-methylbutane chilled with liquid nitrogen and stored at −80 °C. Frozen lung slices with a thickness of 350 μm were sectioned using a HM-560 cryostat (Microm, Heidelberg, Germany) and placed in PBS for fast thawing. A solution consisting of 8 mM CHAPS (ICN biomedicals Inc., Aurora, OH, US), 1 M NaCl and 25 mM EDTA in PBS, pH 8.0, was added to the slices. The solution, 1 mL/slice, was changed six times during a 4 h incubation with mild agitation at room temperature. Additional DNA removal was performed with 30 min incubation of 90 U/mL of benzonase nuclease (Sigma-Aldrich, St Louis, MO, USA) in 20 mM Tris-HCl, 2 mM Mg^2+^ and 20 mM NaCl at pH 8 at 37 °C (1 mL/slice). The slices were then washed in PBS and stored at 4 °C in PBS supplemented with antibiotics (1% PEST, 0.5% gentamicin and 5 μg/mL amphotericin B) [20]. Decellularized lung slices were placed in 12-well plates, 1 slice/well and prewarmed for 1 h in 37 °C in 2 mL/well of DMEM (Sigma-Aldrich, St Louis, MO, USA) supplemented with 10% FCIII (Thermo Scientific, Waltham, MA, USA), 1% PEST and 2 mM L-glutamine. After the incubation, the medium was replaced by 2 mL of fresh medium and 0.195 × 10^6^ HFL-1 cells were added to each lung scaffold slice. Scaffolds with fibroblasts were placed on an orbital shaker (100 rpm, 19 mm orbit) and incubated for 6 h at 37 °C and 5% CO_2_ before LAD2 mast cells (0.15 × 10^6^ /scaffold) were added and incubation continued overnight. The following day, each scaffold containing fibroblasts or co-cultures of fibroblasts and LAD2 were strapped to holders and transferred to new wells containing fresh medium.

### 4.4. Stimulations with Proteases in Primary Lung Fibroblast Cultures

Primary lung fibroblasts from healthy individuals or patients with IPF in monoculture or in co-culture with LAD2 in 6-well plates were cultured for 72 h, followed by 24 h of starvation in DMEM containing 0.4% FCIII before addition of fresh DMEM containing 0.4% FCIII with or without chymase (1 ng/mL), tryptase (75 ng/mL) or a combination of chymase (1 ng/mL) and tryptase (75 ng/mL) (both from Sigma-Aldrich, St Louis, MO, USA). The chosen concentrations of chymase and tryptase had previously been evaluated [19]. All incubations were made at 37 °C with 5% CO_2_. Cell supernatants were collected after 72 h and stored at −20 °C.

### 4.5. Measurement of Cell Metabolism in Protease Exposed Fibroblast Cultures

Cell metabolism was measured by removing the supernatant and then incubating the lung fibroblasts with dissolved Tetrazolium salt (WST-1) (Roche, Mannheim, Germany) for 1 h. The analysis was performed according to the manufacturer’s instructions.

### 4.6. Immunocytochemistry Staining of Cell Cultures and Lung Scaffolds

Monocultures of HFL-1 and co-cultures of HFL-1 and LAD2 cell were seeded in 4-well chamber slides (154526; Thermo Scientific, Waltham, MA, USA). Seeding densities for HFL-1 were 4600 cells/cm^2^ and 3500 cells/cm^2^ for LAD2 cells in co-cultures. Cells in chamber slides were fixed in 4% formaldehyde and stored in PBS at 4 °C until further analysis. Formalin fixed cells were washed with PBS and permeabilized with 0.2% Triton-X100 (Merck, Darmstadt, Germany), followed by washing twice in tris-buffered saline (TBS). Cells were incubated for 60 min at RT with monoclonal tryptase antibody (M7052, Dako, Glostrup, Denmark), α-SMA mouse monoclonal antibody (C6198, Sigma Aldrich, St Louis, MO, USA) and Phalloidin-iFluor 488 Reagent (ab176753, Abcam, Cambridge, UK). The cells were then washed in TBS and incubated for 45 min with secondary antibodies, goat anti-mouse IgG1 (Alexa Fluor^®^ 647, A21240, Thermo Fisher Scientific, Waltham, MA. USA) followed by washing steps in TBS. Nuclei were stained by using DAPI-containing mounting medium (Dako, Glostrup, Denmark). Cells were imaged using two different microscopy methods. The VS120 slide scanner with XV image processor L100 VS-ASW (Olympus, Tokyo, Japan) and Image viewer software VS-OlyVIA (version 2.9) (Olympus Soft Imaging Solutions GmbH; Münster, Germany) was used for image visualization of the α-SMA staining. Images were acquired and processed with NIS-elements, version: 4.60.02, (Laboratory Imaging, Nikon, Tokyo, Japan). The scaffolds were fixed in 4% formaldehyde for 30 min, washed in PBS and stored at 4°C until further analysis. Fixed repopulated human lung scaffolds, remaining in their holders, were gently washed with PBS, and treated with 0.2% Triton-X (in TBS), followed by washing with TBS. The scaffolds were incubated overnight in 4 °C, with monoclonal tryptase antibody (Dako, Glostrup, Denmark) and α-SMA monoclonal antibody (C6198, Sigma-Aldrich, St Louis, MO, USA). The following day, the scaffolds were washed in TBS and secondary antibody (goat anti-mouse IgG1 Alexa Fluor^®^ 647, A21240, Thermo Fisher Scientific, Waltham, MA. USA) were added to the scaffolds and incubated for 48 h in 4°C. After the incubation time, the scaffolds were washed four times with TBS and the holders were removed. The scaffolds were then gently removed to a special petri dish with a cover glass on the bottom. Cell nuclei were stained using DAPI-containing mounting medium (Dako, Glostrup, Denmark). The lung scaffolds were visualized on a Nikon A1+ confocal microscope with a 20× Plan Apo objective (NA 0.75) or a 60× Apo DIC oil immersion objective (NA 1.40) (Nikon Instruments Inc.). Images were acquired and processed with NIS-elements, version 4.60.02, (Laboratory Imaging, Nikon, Tokyo, Japan).

### 4.7. Scanning Electron Microscopy (SEM) of Repopulated Lung Scaffolds

The repopulated lung scaffolds were fixed in buffer containing 0.1 M Sorensen’s phosphate buffer pH 7.4, 1.5% formaldehyde and 1.5% glutaraldehyde at RT for 2 h. Afterwards, the samples were washed twice in 0.1 M Sorensen ’s phosphate buffer, pH 7.4, followed by dehydration in ethanol. The samples were mounted and examined in a Jeol JSM-7800F FEG-SEM at Lund University Bioimaging Centre (LBIC).

### 4.8. Multiplex Analysis and ELISA

The release of different mediators; vascular endothelial growth factor (VEGF)-A_165_, VEGF-C, hepatocyte growth factor (HGF), fibroblast growth factor 2 (FGF2), interleukin 6 (IL-6), interleukin 8 (IL-8), interleukin 1β (IL-1β), matrix metalloproteinase 9 (MMP-9), stem cell factor (SCF) and c-KIT was measured in the collected supernatants from lung fibroblasts and mast cells and quantified by magnetic bead-based multiplex assay performed according to the manufacturers’ instructions (Luminex Human Magnetic Assay 10-Plex, 96 well, B-Bioplex, Biotechne, Minneapolis, MN, USA). The multiplex analyses were performed by IQ Biotechnology Platform, Lund university, Sweden. The release of IL-6, VEGF-A_165_ and HGF were further measured in cell supernatants by ELISA performed according to the manufacturers’ instructions (IL-6 ELISA assay RAB 0306, Sigma-Aldrich, St Louis, MO, USA, Human VEGF Quantikine ELISA Kit, DVE00, R&D Systems, Minneapolis, MN, USA and Human HGF Quantikine ELISA Kit DHG00, R&D Systems, Minneapolis, MN, USA).

### 4.9. Protein Quantification and Western Blot Analysis

Proteins from cell cultures in 6-well plates were collected by addition of lysis buffer RIPA (89900, Pierce^TM^, Thermo Fisher Scientific, Waltham, MA. USA) (100 μL/well) containing a protease inhibitor (11697498001, Complete, Roche) and centrifugation at 14,000× *g* for 15 min. The lysate was stored at −20 °C until further analysis. Protein content was determined using a BCA assay kit (23225, Thermo Fisher Scientific, Waltham, MA. USA). For Western blot analysis, gel electrophoresis was used in order to size-separate the proteins on a Mini-Protean TGX stain-free gel 7.5% (456-8026, Bio-Rad, Hercules, CA, USA). Afterwards, the proteins were transferred to a Trans Blot Turbo 0.2 μm LF PVDF membrane (170-4274, Bio-Rad, Hercules, CA, USA), followed by a blocking step with 5% BSA (A2153, Sigma-Aldrich St Louis, MO, USA) in TBS-Tween (0.05%) (170-6435, Biorad, Solna, Sweden; P1379, Sigma-Aldrich St Louis, MO, USA). The primary antibody was added, and the membrane incubated (overnight at 4 °C); α-SMA polyclonal antibody (1:1000, ab5694, Abcam, Cambridge, UK). The membranes were washed several times in TBS-Tween and incubated with the secondary antibody goat anti-rabbit IgG-HRP conjugated (1:5000, 170-6515, Biorad, Solna, Sweden) and Precision Protein™ StrepTactin-HRP Conjugate (1:7000, 1610381, Biorad, Solna, Sweden). Following four washing steps, the targeted protein bands were visualized (Electrochemiluminescence ECL Clarity Western ECL (Biorad, Solna, Sweden) and quantified with ChemiDoc™ Touch Imaging System (Biorad, Solna, Sweden).

### 4.10. Migration of Human Lung Fibroblasts and Alveolar Epithelial Cells

Migration of HFL-1 was performed as previously described [19]. The epithelial cell line A549 was seeded in 24-well plates with a cell density of 30,000 cells/well and became confluent after 24 h in culture media. After starvation for 20 h, the cell layer was scratched with a 200 µL tip to create a scratch and then exposed for 24 h to different concentrations of IL-6 (0.3, 1 and 3 ng/mL), VEGF (0.5, 1.5 and 15 ng/mL), HGF (5 and 10 ng/mL) or conditioned medium, collected after 72 h from the cell cultures on plastic plates or lung scaffold cultures. Matching internal controls for the individual experiments were always included and compared in the statistical analysis. Live imaging was performed and changes in scratch area were measured with NIS Elements AR Analysis software after 12 h and 24 h and expressed as percentage of change in scratch area compared to the starting point 0 h.

### 4.11. Statistical Methods

Statistical analyses and graphs were generated using the GraphPad software (GraphPad Software Prism 7, La Jolla, CA, USA). Student’s (paired) *t*-test was used for two group comparisons and one-way or two-way repeated measures ANOVA or one-sample *t*-test were used for relevant comparisons with more than two groups in the different experiments. *P*-values of *p* < 0.05 were considered as statistically significant.

## Figures and Tables

**Figure 1 ijms-22-00506-f001:**
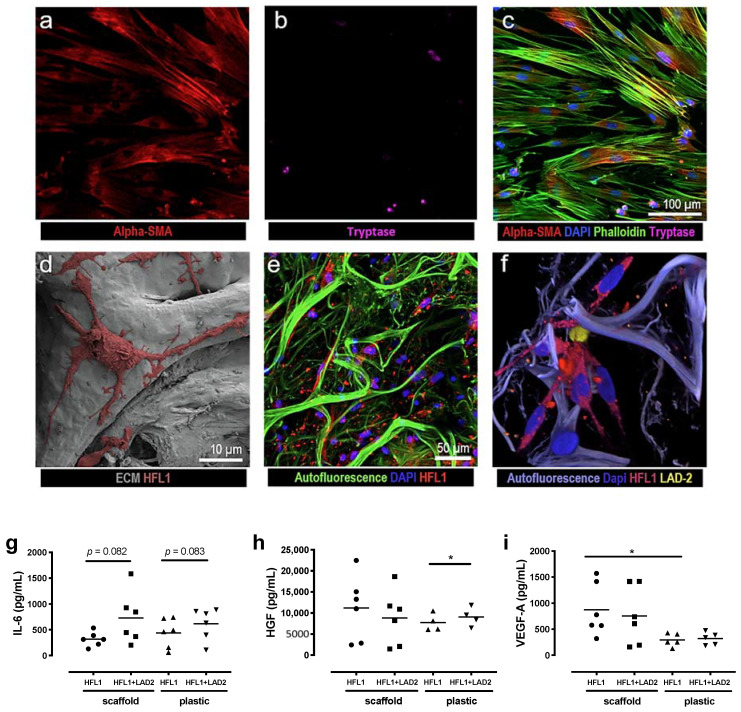
Representative images and mediator release from culture of human fetal lung fibroblasts (HFL-1) and LAD2 cells on cell culture plastic plates and in 3D lung scaffolds with extracellular matrix (ECM) matrices. Top panel shows images from co-cultures of HFL-1 and LAD2 on plastic plates visualized with confocal microscopy (**a**–**c**) where (**a**) fibroblasts are stained for α-smooth muscle actin (SMA) and (**b**) LAD2 cells have been stained for tryptase, (**c**) shows a merged image with additional staining for actin with phalloidin and cell nuclei with DAPI. Bottom panel shows cultures in (**d**) 3D decellularized ECM matrices with scanning electron microscope (SEM) images of repopulating HFL-1 cells. (**e**) Confocal microscopy images of HFL-1 cells labelled with cytopainter and (**f**) a co-culture of HFL-1 (red) and LAD2 cells (yellow) both labelled with cytopainter. (**g**–**i**) Mediator release from cells cultured in cell culture plastic plates or lung scaffolds. (**g**) IL-6, (**h**) hepatocyte growth factor (HGF) and (**i**) vascular endothelial growth factor (VEGF) were quantified in cell culture medium after 72 h in fibroblast cultures with and without mast cells in cell culture plastic plates or lung scaffolds. Data are presented as individual values with means, *n* = 2 or 3 individual experiments with two technical replicates. Statistical analyses were performed with Student’s (paired) *t*-test. * *p* < 0.05.

**Figure 2 ijms-22-00506-f002:**
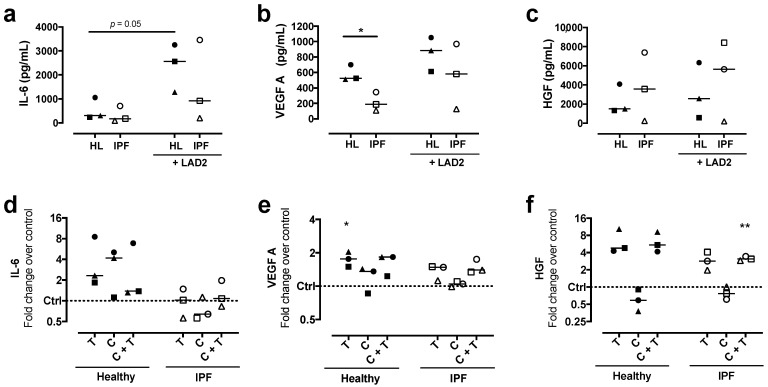
Mediator release from healthy and idiopathic pulmonary fibrosis (IPF) fibroblasts. Multiplex cytokine analysis of the cell culture supernatant was performed for IL-6, vascular endothelial growth factor (VEGF)-A_165_ and hepatocyte growth factor (HGF) from healthy (HL) and IPF (IPF) fibroblasts in monocultures and in co-cultures with LAD2 mast cells (**a**–**c**). Healthy and IPF fibroblasts were stimulated for 72 h with either tryptase 75 ng/mL (T), chymase 1 ng/mL (C) or a combination of tryptase and chymase (T + C). Data is presented as fold change compared to unstimulated control (**d**–**f**) with individual values and median. *n* = 3 for each group with two pooled technical replicates in each experiment, represented as one data point. Statistical analyses were performed with unpaired or paired Student’s *t*-test for the monoculture versus co-culture comparisons and one-sample *t*-test was performed for protease comparisons versus untreated control. * *p* < 0.05; ** *p* < 0.01.

**Figure 3 ijms-22-00506-f003:**
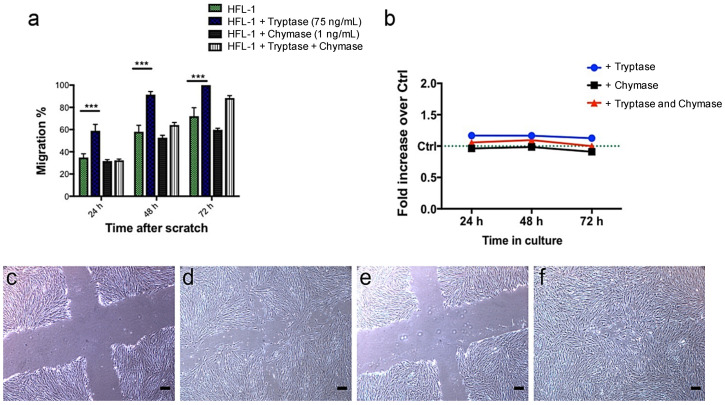
Tryptase has a promigratory effect on human fetal lung fibroblasts (HFL-1) in a scratch assay without altering overall metabolic activity. HFL-1 cells were stimulated for 72 h with tryptase (75 ng/mL), chymase (1 ng/mL) or a combination of tryptase and chymase. The migratory capacity of HFL-1 cells was measured at 24 h, 48 h and 72 h as the percentage of cell-occupied space compared to time 0 h when the scratch was made. (**a**) Tryptase increased the migratory capacity of HFL-1, while chymase had no significant effect (*n* = 3). Tryptase and chymase together did not show a statistically significant promigratory effect. Data are presented as mean ± SEM, and statistical analysis was performed with Student’s *t*-test. *** *p* < 0.001 (**b**) The metabolic activity of HFL-1 was measured by the WST-1 assay at three different time points (24, 48 and 72 h) and presented as changes in fold increase versus control. There was no apparent effect of tryptase, chymase or the combination of tryptase and chymase on the metabolic activity of HFL-1. (**c**–**f**) Representative images from the migration experiments (**c**) HFL-1 (no stimuli added) at starting points 0 h and (**d**) 48 h, (**e**) HFL-1 + Tryptase at starting points 0 h and (**f**) 48 h. Scale bars = 100 µm.

**Figure 4 ijms-22-00506-f004:**
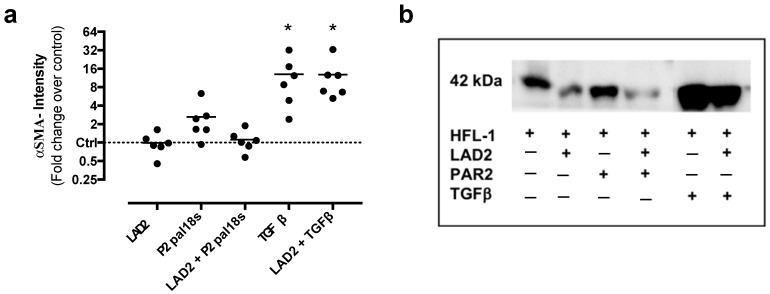
Protein expression of α-smooth muscle actin (SMA) was not altered by LAD2 mast cells or protease-activated receptor 2 (PAR2) antagonist. (**a**) Human fetal lung fibroblasts (HFL-1) in monocultures and in co-cultures with LAD2 mast cells were treated for 72 h with the PAR2 antagonist P2pal-18s (10 µM). Cell cultures treated with transforming growth factor (TGF)-β (10 ng/mL) were included as positive controls. Cell lysates were collected and protein levels of α-SMA were analyzed with Western blot. (**b**) Representative protein bands are shown for α-SMA. PAR2 = PAR2 antagonist P2pal-18s. Data are presented as mean values of fold change of α-SMA intensity versus untreated control and normalized against the total protein for each lane. *n* = 6 independent experiments with pooled duplicates in each data point (**a**). Statistical analyses were performed with one-sample *t*-test versus untreated control. * *p* < 0.05.

**Figure 5 ijms-22-00506-f005:**
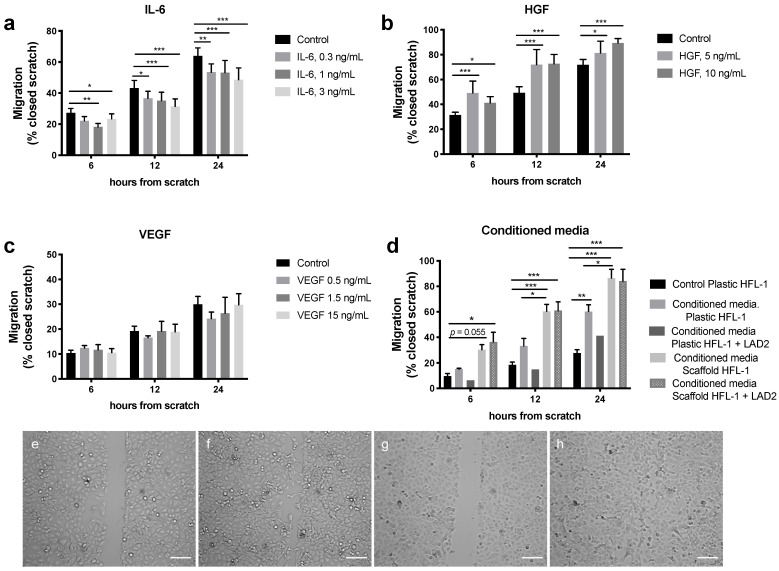
Migratory effect on epithelial A549 cells after exposure to (**a**) IL-6 (0.3, 1 and 3 ng/mL), (**b**) Hepatocyte growth factor (HGF) (5 and 10 ng/mL), (**c**) Vascular endothelial growth factor (VEGF) (0.5, 1.5 and 15 ng/mL) or (**d**) conditioned media from fibroblasts and LAD-2 mast cells cultured on either plastic culture plates or in lung scaffolds compared to control (A549 cells without stimuli). (**e**–**h**) Example images of the scratch experiments (**e**) Control (no stimuli added) at starting point 0 h, (**f**) control at 24 h, (**g**) HGF at starting point 0 h, (**h**) HGF stimuli at 24 h. Migration of epithelial cells was measured as percentage of closed scratch at 6 h, 12 h and 24 h compared to the starting point 0 h. Data are presented as mean ± SEM (*n* = 3–12, except for conditioned media scaffolds HFL-1 + LAD2 were *n* = 2, which was not included in the statistical analyses). Statistical analyses were performed with two-way repeated measures ANOVA. * *p* < 0.05; ** *p* < 0.01; *** *p* < 0.001. Scale bars = 100 µm.

**Figure 6 ijms-22-00506-f006:**
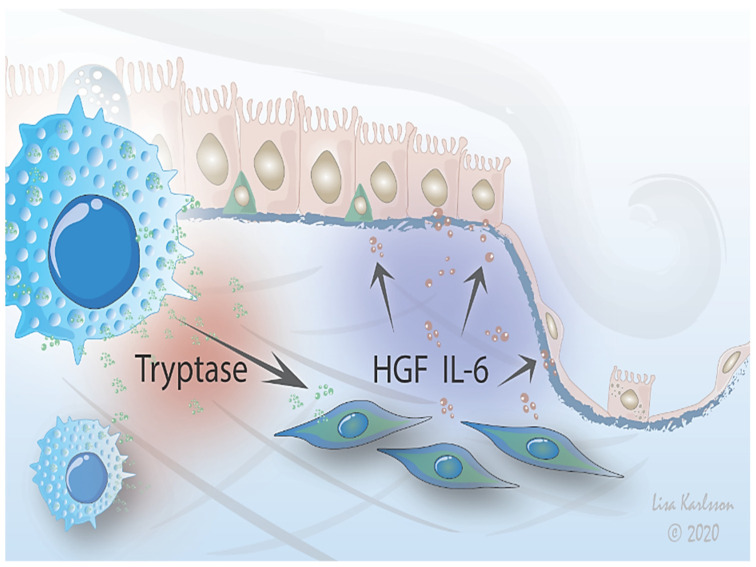
Proposed mechanism. A stimulus is triggering crosstalk between mast cells and lung fibroblasts that influences migration of alveolar epithelium. Mast cells interact with fibroblasts which in turn signals to epithelial cells via IL-6 and HGF signaling, affecting the epithelial response to injury by either promoting resolution or pathological remodeling. The interactions between mast cells and fibroblasts are further modified by the ECM environment.

## Data Availability

The data will be available upon request.

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
