# Peer review of "Crosstalk between Mast Cells and Lung Fibroblasts Is Modified by Alveolar Extracellular Matrix and Influences Epithelial Migration"

_ijms, 2021, doi:10.3390/ijms22020506_

Round 1

Reviewer 1 Report

The authors investigated the crosstalk between mast cells and fibroblasts in the presence or not of extracellular matrix. Mast cell tryptase increases the release of IL-6, VEGF, and HGF by fibroblasts whereas the presence of mast cells induces only IL-6 release. The presence of extracellular matrix increases the release of VEGF and HGF. Migration of epithelial cells was reduced by IL-6 but increased by HGF.

This is an interesting paper but the conclusions are too strong for the data and more experiments and controls are needed. Given the variability of the data, the number of experiments has to be increased to draw any conclusions in figure 1 and 2. Figure 1g, h and i represent 2 experiments made in duplicate. No statistic can be done with n=2. Given the variation in data and that this is co-culture of cell lines, there is no reason for not doing more experiments. Furthermore, there is one control missing, the mast cells alone. Mast cells are well known to produce both IL-6 and VEGF. The increase of IL-6 (Fig 1) and VEGF (Fig 2) observed in presence of mast cells may come from the mast cells and not from fibroblasts. Intracellular staining is required to draw any conclusion.

The title is on the crosstalk between mast cells and fibroblasts but in figure 4, the authors did not use the supernatants of co-cultured mast cells and fibroblasts, they used only fibroblasts. The effects of supernatants of mast cells alone and co-cultured mast cells and fibroblasts are needed.

In summary, this paper is incomplete and needs further experiments and controls.

Minor comments:

Line 193: abbreviation a-SMA

Line 223: abbreviation TGFb

Line 293 and 302: reference Rosmark should be number 20

Line 360: abbreviation IL-1b

Author Response

Reviewer 1:

We would like to thank the reviewer for valuable comments that have helped us to improve research design, method descriptions, presentation of the results and conclusions in the revised manuscript. We have also performed spell check for English language and style.

#1 This is an interesting paper but the conclusions are too strong for the data and more experiments and controls are needed. Given the variability of the data, the number of experiments has to be increased to draw any conclusions in figure 1 and 2. Figure 1g, h and i represent 2 experiments made in duplicate. No statistic can be done with n=2. Given the variation in data and that this is co-culture of cell lines, there is no reason for not doing more experiments.

Reply #1 We understand the concerns raised by the reviewer. We presented in figure 1 data from the samples we used for the migration experiments on epithelial cells. Unfortunately, there were not enough samples left from the remaining two experimental settings for additional migration experiments. We have now updated Figure 1 on page 4 with the remaining data from the two culture conditions, scaffolds and plastic (in total n=3 with 2 technical replicates for each culture conditions) and also updated the result section on page 3, lines 108-128 and the following discussion section on page 9 and lines 407-410.

No statistical comparisons were done for the data with n=2. We have re-analyzed the multiplex data set and found that existing data from one of the IPF patients had not been reported for IL-6 (fig 2d) and HGF (fig 2c and f). This was due to a lapse in communication with a former group member. Figure 2 is now updated on page 5 with data and statistical analysis from all the IPF fibroblasts (n=3). The result section on page 5 is also updated.

#2 Furthermore, there is one control missing, the mast cells alone. Mast cells are well known to produce both IL-6 and VEGF. The increase of IL-6 (Fig 1) and VEGF (Fig 2) observed in presence of mast cells may come from the mast cells and not from fibroblasts. Intracellular staining is required to draw any conclusion.

Reply #2 This is an important control which we now have included in the manuscript. Samples from mono-cultured LAD2 were also analysed in our experiments presented in fig 1 and 2. We have measured the release of IL-6, VEGF and HGF from LAD2 both with ELISA (figure 1) and multiplex (figure 2). These data are now presented in the result section on page 3, lines 108-124: We analyzed the release of IL-6, VEGF and HGF by Enzyme-Linked ImmunoSorbent Assay (ELISA) in the two culture conditions using HFL-1 fibroblasts and LAD-2 mast cells (Figure 1a-f). The LAD2 cells in these experiments were not stimulated to induce degranulation. Mast cells alone released low amounts of IL-6 (12.62 ± 1.62 pg/mL) in cell culture plastic plates and no detectable levels in scaffold cultures. Fibroblasts were the major producers of IL-6. ….” “Mast cells in mono-culture released low amounts of HGF (19.19 pg/mL) in cell culture plastic plates and in scaffold cultures (30.61 ± 1.59 pg/mL). Fibroblasts were the main producers of HGF”… “Mast cells in mono-culture did not release detectable levels of VEGF whereas low amounts of VEGF were released from the mast cells in scaffold cultures (16.85 ± 1.59 pg/mL). Fibroblasts were the major producers of VEGF.” And on page 5, lines 186-188: “Mast cells in mono-culture released IL-6 (2.08 ± 0.17), VEGF-A (89.39 ± 3.5), and no detectable levels of HGF in the multiplex analyses”. and in the discussion section on page 10, lines 448-456: In the present study we investigated if cellular contact between mast cells and fibroblast altered fibroblast activity by release of different mediators. Importantly, the mast cells were not activated in terms of degranulation to secrete inflammatory mediators and we did not detect any major release of the investigated mediators VEGF, IL-6 or HGF. Mast cells are otherwise well known to release both IL-6 and VEGF upon activation and degranulation [36]. These data support that fibroblasts were the major producers of the measured mediators which were further increased in cellular contact with mast cells.”

#3 The title is on the crosstalk between mast cells and fibroblasts but in figure 4, the authors did not use the supernatants of co-cultured mast cells and fibroblasts, they used only fibroblasts. The effects of supernatants of mast cells alone and co-cultured mast cells and fibroblasts are needed.

Reply#3 We have included also the data from fibroblasts co-cultured with LAD2 in figure 4, now updated to Figure 5 on page 8. We did not have any data from mast cells alone in this experimental set-up. The result section on page 7-8.

#4 In summary, this paper is incomplete and needs further experiments and controls.

Reply #4 We have added data in figure 1, 2 and 5 and mast cell controls in the result sections on page 3-8 and a new figure 4 with a-SMA data.

Minor comments:

#5 Line 193: abbreviation a-SMA

Reply #5 We have now corrected to a-SMA

#6 Line 223: abbreviation TGFb

Reply #6 We have now corrected to TGFb

#7 Line 293 and 302: reference Rosmark should be number 20

Reply #7 The reference is now presented correctly

#8 Line 360: abbreviation IL-1b

Reply #8 The abbreviation is now corrected IL-1b

Reviewer 2 Report

The manuscript entitled "Crosstalk between mast cells and lung fibroblasts is modified by alveolar extracellular matrix and influences epithelial migration" by Bagher and colleagues details the results of several experiments designed to investigate the role of mast cell - lung fibroblast crosstalk on cell signaling that influences alveolar epithelial behavior in the context of idiopathic pulmonary fibrosis. Data from several experiments are presented utilizing variations of in vitro studies with different combinations of cells and conditions. Data presented from a human lung fibroblast cell line (HLF-1) and mast cell line (LAD2) demonstrate that cytokine production is altered depending of co-culture and the culture methods used (scaffold vs plastic). The authors go on to show that there are some differences in cytokine profiles in the HLF/MC co-cultures if the HLFs are derived from healthy vs IPF donors. Also suggested is that cytokine production can be influenced by addition of mast cell proteases instead of co-culture with LAD2 cells. The authors go on to demonstrate that mast cell proteases alter the migration of HLF-1 cells without altering their metabolic state. The authors then shift to a model of alveolar epithelial cells (A549) and show that the altered cytokine profiles from the mast cell (or mast cell protease)/HLF co-culture experiments affect the migration of A549 cells. They summarize their findings the final figure. The text agrees with the data that is presented. This manuscript is of significant interest beyond the IPF realm and would be of interest to investigators studying mast cell, fibroblast interactions in other areas. On the whole, the manuscript is well written and well edited; however, I do have the following concerns with the manuscript in its present form:

Major concerns:

1) Figure 1 - The title of the figure implies that the main point of the figure is to report differences in "mediator release" from the different cell culture methods. While the microscopy is gorgeous, it does not really add to this data. Perhaps a the figure could be reconstructed to emphasize the comparison data first and then utilize the microscopy to to highlight differences between the 2D and 3D culture models? Also the cytokines data as presented is a bit confusing. The authors report that the data is the mean of 2 separate experiments with 2-3 technical replicates, yet there are 4 data points for each group. This is confusing and needs clarification. Also these data are obtained from cell lines so I'm curious as to why the authors stopped at 4 data points given that there is such high variability within groups. For example, groups in panel G demonstrate a wide bimodal distribution, which is clearly driving the effect.

2) Figure 2 - The authors explain in the Discussion that primary HLF samples were limited, which explains the small N in this dataset. There is significant splay in the data for the cultures in Panel A and B for the IPF groups (N=3, not surprising for primary cells). The remaining panels of this figure only show data from N=2 IPF cell lines (except Panel E and one group in Panel D). Was data from the 3rd line excluded as an outlier or were the IPF samples so limited that only they only ran the assay on some reported experiments? The unbalanced groups in Panel D further raise concern for missing data. This should be addressed up front in the Results as making comparisons for N=2 data is problematic. If outlier data was excluded, this should be discussed.

3) Figure 4 - The panels in this figure show the results of experiments using A549 cells in a scratch migration assay with supplementation of the cytokines shown to be altered in the MC/HLF experiments. Given that this is a cell line, I'm concerned about the differences in control data across the experiments. For example the untreated A549 migration percentage is ~75% at 24 hours for the data presented in Panels A & B. However, the control data at 24 hours in Panels C & D for the A549 migration is closer to 35%. This is a big discrepancy given that this data is obtained from a cell line and suggests that there may be other experimental differences other than the cytokines or conditioned media in these studies. These discrepancies among the control data need to be addressed.

4) Alpha-SMA Data (Discussion) - The authors state that they could not establish differences in fibroblast to myofibroblast transition based on measurements of alpha-SMA (data not shown). How were these data measured (PCR, IF, Flow)? Since this data is alluded to, but not shown, it would be important to know how this was assessed. This would be of interest given the profibrotic effects of tryplase that the authors mention in the sentence prior.

Minor concerns:

1) The authors may consider showing representative images for Figure 3a similar to Figure 4.

2) There are several instances throughout the manuscript where the special characters such as the greek letters for alpha and beta are not showing appropriately and need to be corrected.

Author Response

Reviewer 2:

We would like to thank the reviewer for valuable comments that have helped us to improve research design, method descriptions, presentation of the results and conclusions in the revised manuscript. We have also performed spell check for English language and style.

The manuscript entitled "Crosstalk between mast cells and lung fibroblasts is modified by alveolar extracellular matrix and influences epithelial migration" by Bagher and colleagues details the results of several experiments designed to investigate the role of mast cell - lung fibroblast crosstalk on cell signaling that influences alveolar epithelial behavior in the context of idiopathic pulmonary fibrosis. Data from several experiments are presented utilizing variations of in vitro studies with different combinations of cells and conditions. Data presented from a human lung fibroblast cell line (HLF-1) and mast cell line (LAD2) demonstrate that cytokine production is altered depending of co-culture and the culture methods used (scaffold vs plastic). The authors go on to show that there are some differences in cytokine profiles in the HLF/MC co-cultures if the HLFs are derived from healthy vs IPF donors. Also suggested is that cytokine production can be influenced by addition of mast cell proteases instead of co-culture with LAD2 cells. The authors go on to demonstrate that mast cell proteases alter the migration of HLF-1 cells without altering their metabolic state. The authors then shift to a model of alveolar epithelial cells (A549) and show that the altered cytokine profiles from the mast cell (or mast cell protease)/HLF co-culture experiments affect the migration of A549 cells. They summarize their findings the final figure. The text agrees with the data that is presented. This manuscript is of significant interest beyond the IPF realm and would be of interest to investigators studying mast cell, fibroblast interactions in other areas. On the whole, the manuscript is well written and well edited; however, I do have the following concerns with the manuscript in its present form:

Major concerns:

#1 Figure 1 - The title of the figure implies that the main point of the figure is to report differences in "mediator release" from the different cell culture methods. While the microscopy is gorgeous, it does not really add to this data. Perhaps a the figure could be reconstructed to emphasize the comparison data first and then utilize the microscopy to to highlight differences between the 2D and 3D culture models? Also the cytokines data as presented is a bit confusing. The authors report that the data is the mean of 2 separate experiments with 2-3 technical replicates, yet there are 4 data points for each group. This is confusing and needs clarification. Also these data are obtained from cell lines so I'm curious as to why the authors stopped at 4 data points given that there is such high variability within groups. For example, groups in panel G demonstrate a wide bimodal distribution, which is clearly driving the effect.

Reply #1 We thank the reviewer for valuable suggestion. We decided to keep figure 1 as it is to first present the two models and then data. As presented in the reply #1 to reviewer 1, We understand the concerns raised by both the reviewers. We presented in figure 1 the data from the samples we used for the migration experiments on epithelia cells. Unfortunately, there were not enough samples left from the remaining two experimental settings for additional migration experiments. We have now updated Figure 1 on page 4 with the remaining data from the two culture conditions, scaffolds and plastic (in total n=3 with 2 technical replicates for each culture conditions) and also updated the result section on page 3, lines 108-128 and the following discussion section on page 9 and lines 407-410.

#2 Figure 2 - The authors explain in the Discussion that primary HLF samples were limited, which explains the small N in this dataset. There is significant splay in the data for the cultures in Panel A and B for the IPF groups (N=3, not surprising for primary cells). The remaining panels of this figure only show data from N=2 IPF cell lines (except Panel E and one group in Panel D). Was data from the 3rd line excluded as an outlier or were the IPF samples so limited that only they only ran the assay on some reported experiments? The unbalanced groups in Panel D further raise concern for missing data. This should be addressed up front in the Results as making comparisons for N=2 data is problematic. If outlier data was excluded, this should be discussed.

 Reply #2 We agree with the reviewer and as we also replied #1 to reviewer 1: We have now re-analysed the multiplex data set to understand why this data was not presented. We found that existing results from one of the IPF patients had not been reported for IL-6 (fig 2d) and HGF (fig 2c and f). This was due to a lapse in communication with a former group member. Figure 2 is now updated on page 5 with measured data and statistical analysis from the IPF fibroblasts (n=3). The result section on page 5 is also updated.

#3 Figure 4 - The panels in this figure show the results of experiments using A549 cells in a scratch migration assay with supplementation of the cytokines shown to be altered in the MC/HLF experiments. Given that this is a cell line, I'm concerned about the differences in control data across the experiments. For example, the untreated A549 migration percentage is ~75% at 24 hours for the data presented in Panels A & B. However, the control data at 24 hours in Panels C & D for the A549 migration is closer to 35%. This is a big discrepancy given that this data is obtained from a cell line and suggests that there may be other experimental differences other than the cytokines or conditioned media in these studies. These discrepancies among the control data need to be addressed.

Reply #3 The migration experiments were carefully designed, although differences in migratory capacity were observed between the different epithelial migration experiments, especially for the VEGF experiments. We have added in the method section on page 15, lines 681-683: “Matching internal controls for the individual experiments were always included and compared in the statistical analysis”.

#4 Alpha-SMA Data (Discussion) - The authors state that they could not establish differences in fibroblast to myofibroblast transition based on measurements of alpha-SMA (data not shown). How were these data measured (PCR, IF, Flow)? Since this data is alluded to, but not shown, it would be important to know how this was assessed. This would be of interest given the profibrotic effects of tryplase that the authors mention in the sentence prior.

Reply #4 We decided to present the result from the a-SMA measurements which were done by western blots (Figure 4 on page). We also included the PAR2 antagonist to and the positive control TGFb. We have updated the method section with 4.9. Protein quantification and western blot analysis on page 14, lines 651-673, result section 2.4 2.4. Effects on aSMA expression after co-culture with mast cells on page 6, lines 303-329 and discussion section on page 9, lines 380-384.

Minor concerns:

#1 The authors may consider showing representative images for Figure 3a similar to Figure 4.

Reply #1 We have now added representative images for fibroblast migration in Figure 3 (c-f) on page 6.

#2 There are several instances throughout the manuscript where the special characters such as the greek letters for alpha and beta are not showing appropriately and need to be corrected.

Reply #2 We have now corrected these issues in the manuscript

Reviewer 3 Report

In this manuscript the authors describe their findings regarding the role of mast cells in influencing the production of cytokines by human lung fibroblasts in the context of UIP/IPF. 

The paper is well-written and the results appear convincing. Please provide p-values for the panels in the legend of figure 2. 

Author Response

#1 The paper is well-written and the results appear convincing. Please provide p-values for the panels in the legend of figure 2. 

Reply #1: We have now included the p-values in the legend for figure 2

Round 2

Reviewer 1 Report

The authors responded to some of the comments. They increased the number of experiments in figure 1 which changed the conclusion. There was no significant increase of IL-6 in presence of mast cells and probably no increase of HGF. They should increase the n of the experiment on the release of HGF and verify their statistic because it does not seem to have a significant difference.

Although mast cells were not stimulated to degranulate, the presence of fibroblasts could activate them to release IL-6 without degranulation. Thus, identification of the cells producing IL-6 is required. Furthermore, what will happen when mast cells are stimulated?

In the summary line 32 and in the discussion lines 295-296, it is written that co-culture of fibroblasts with mast cells increase the release of IL-6, but the data do not show that in figure 1. Although, there is an increase of IL-6 in figure 2. How do the authors explain this discrepancy? Do fetal fibroblasts act differently than cell line fibroblasts? This should be discussed.

Reviewer 2 Report

The revision of the manuscript by Bagher and colleagues entitled "Crosstalk between mast cells and lung fibroblasts is modified by alveolar extracellular matrix and influences epithelial migration" represent a substantial revision of the previously submitted manuscript version. The author's have added additional data to satisfy queries by myself and the other Reviewer's. Overall, I believe that the manuscript is significantly improved and adequately addresses my comments on the prior manuscript version. I have the following comments on the present draft.

(New) Figure 4 - This is a new figure added to the manuscript to demonstrate data that was previously eluded to in the discussion only. In the present form the structure of the figure could be improved. It is not clear to me why the data are broken out into panels A-C. Statistical differences are reported in panel C relative baseline data (shown in panel A). It is unusual to show comparison data in separate figure panels. The comparisons would be more appropriate if panels A, B, and C were combined into a single panel. Also the significance notation in panel C is difficult to read in the present figure.
